# Investigating Farm Fragmentation as a Risk Factor for Bovine Tuberculosis in Cattle Herds: A Matched Case-Control Study from Northern Ireland

**DOI:** 10.3390/pathogens11030299

**Published:** 2022-02-26

**Authors:** Georgina Milne, Jordon Graham, John McGrath, Raymond Kirke, Wilma McMaster, Andrew William Byrne

**Affiliations:** 1Agri-Food and Biosciences Institute (AFBI), Belfast BT4 3SD, UK; jordon.graham@afbini.gov.uk; 2Department of Agriculture, Environment and Rural Affairs (Area Based Scheme), Londonderry BT48 6AT, UK; john.mcgrath@daera-ni.gov.uk; 3Department of Agriculture, Environment and Rural Affairs (Veterinary Service Animal Health), Limavady BT49 9HP, UK; raymond.kirke@daera-ni.gov.uk; 4Department of Agriculture, Environment and Rural Affairs (Land Parcel Identification System), Ballymena BT43 6HY, UK; wilma.mcmaster@daera-ni.gov.uk; 5One-Health Unit, Department of Agriculture, Food and the Marine, D02 WK12 Dublin, Ireland; andreww.byrne@agriculture.gov.ie

**Keywords:** farm fragmentation, bTB, bovine tuberculosis, Northern Ireland, local spread, neighbourhood, matched case-control, conacre

## Abstract

Bovine tuberculosis remains a challenging endemic pathogen of cattle in many parts of the globe. Spatial clustering of *Mycoacterium bovis* molecular types in cattle suggests that local factors are the primary drivers of spread. Northern Ireland’s agricultural landscape is comprised of highly fragmented farms, distributed across spatially discontinuous land parcels, and these highly fragmented farms are thought to facilitate localised spread. We conducted a matched case control study to quantify the risks of bovine tuberculosis breakdown with farm area, farm fragmentation, fragment dispersal, and contact with neighbouring herds. Whilst our results show small but significant increases in breakdown risk associated with each factor, these relationships were strongly confounded with the number of contiguous neighbours with bovine tuberculosis. Our key finding was that every infected neighbour led to an increase in the odds of breakdown by 40% to 50%, and that highly fragmented farms were almost twice as likely to have a bTB positive neighbour compared to nonfragmented farms. Our results suggest that after controlling for herd size, herd type, spatial and temporal factors, farm fragmentation increasingly exposes herds to infection originating from first-order spatial neighbours. Given Northern Ireland’s particularly fragmented landscape, and reliance on short-term leases, our data support the hypothesis that between-herd contiguous spread is a particularly important component of the region’s bovine tuberculosis disease system.

## 1. Introduction

Bovine tuberculosis (bTB), caused primarily by *Mycobacterium bovis* infection, is a complex and challenging disease of cattle, endemic in many countries across the globe [1]. The disease continues to blight the cattle industry in the United Kingdom (UK) and Republic of Ireland (ROI), despite long-running test-and-slaughter eradication programmes [2,3,4], ancillary testing [5,6,7], and surveillance at the abattoir for lesions indicative of bTB [8,9]. Failure to eradicate the disease is in part a consequence of multiple nonmutually exclusive infection pathways, which can add considerable complexity to disease control [10].

Whilst *M. bovis* can be introduced into a herd via processes operating over relatively long distances (for example, the purchasing of cattle from other herds or marts [11,12,13]), the persistence and spread of *M. bovis* in the UK and ROI is understood to be particularly dependent on “local” factors [14,15,16,17] These include (but are not limited to) spillback facilitated by direct or indirect contact with infected wildlife [18,19,20,21], direct contact with neighbouring herds [16,22,23], and a contaminated environment [24]. 

Farm fragmentation, whereby farms are distributed across multiple, spatially discontinuous land parcels, is particularly common on the island of Ireland [25,26]. In Northern Irish cattle farms, 35% of businesses were comprised of five or more fragments [27], comparable to the ROI where 32% of sampled farms were comprised of five or more fragments [28]. In contrast, a study from Great Britain reported that only 7% of sampled farms had five or more constituent fragments [29]. Farm fragmentation may contribute to the bTB epidemic by providing more opportunities for direct nose to nose contact between cattle from other farms “over the fence” [16,22,23]. 

Contiguous cattle-to-cattle spread can be counteracted with robust biosecurity measures such as double fencing between parcels with a 3 m gap between neighbours [30]. There is no guarantee, however, that these measures are enacted. In a study of farms in north-west England, farming units had contact with between one and seventeen neighbouring farms [31]. In three separate NI-based studies, 79% [32] and 67% [12] of fields permitted contact with neighbouring farms, and cattle were found to spend up to 40% of their grazing days beside neighbours [33]. To try and control contiguous spread, surveillance in NI involves lateral check tests. These are bTB tests applied to neighbouring herds which are grazed in proximity to a breakdown herd, and are therefore deemed at risk by veterinary inspectors. Lateral check tests involve intensive mapping exercises and biosecurity assessments, but due to the fragmented nature of cattle farms in NI, the process of identifying at-risk herds and parcels can be challenging.

Despite earlier studies alluding to the potential for fragmented farms to disseminate infection [10,34] this exposure’s contribution to the epidemic in NI remains poorly understood. Furthermore, to the best of the authors’ knowledge, this is the first published research to explicitly explore the risk between farm fragmentation and risk of bTB breakdown within cattle herds. The primary objective of this research, therefore, is to quantify the impact of each of 1) farm fragmentation, 2) fragment dispersal and 3) contiguous contact and 4) farm area, on the odds of bTB breakdown. The secondary objective is to assess for confounding and interaction effects with the presence of neighbours with bTB.

## 2. Results

### 2.1. Descriptive Results

The final dataset consisted of 19,008 herds, from which 4,637 (24.4%) had at least one confirmed bTB breakdown during the three year study period; the spatial distribution of case herds in the whole cattle population is illustrated in Figure 1. These case herds were matched to 4,637 controls that did not have a bTB breakdown between 2015 and 2017, Table 1. The average case farm was 8 ha larger (IQR: 1 ha–12 ha) than the average control farm, and 41% of case farms were classified as “Large” (>59.1 ha)”, compared to 33% of controls. The average case farm was associated with five additional land parcels than the average control farm, and 21.9% of case farms were “highly fragmented” (8–10 fragments) or “very highly fragmented” (11+ fragments)”, compared to 17.7% of control farms. 30% of case farms had very highly dispersed fragments (>3.05 km between fragment centroids), compared to 26% of control farms, and 41% of case farms had the highest levels of contact (>4.94 km of shared boundary) with neighbouring farms, compared to 33% of control farms. Some 71% of case farms had at least one contiguous neighbour with a confirmed bTB breakdown (up to a maximum of 14), compared to a 53% of control farms (up to a maximum of 11). 

### 2.2. Univariable Models

Moderate to strong correlation was observed between farm area, farm fragmentation, fragment dispersal and contact with neighbouring farms (R-Markdown Appendix A), with the strongest correlation between the number of fragments and the extent of shared boundary (ρ = 0.7; Appendix A, Figure 2). The results of the univariable analyses are shown in Table 2. The Locally Estimated Scatterplot Smoothing (LOESS) plot of the relationship between bTB status and farm area is shown in Appendix A, Figure 2A. Whilst no association was observed between risk of bTB breakdown and farm area in the continuous model, the categorical model shows that only the very largest farms (>59.1 ha) were associated with elevated bTB positive status, compared to medium sized farms (16.41 ha–31.2 ha, OR: 1.87; 95% CI: 1.55–2.25), see Figure 2A. A positive relationship can also be observed between the number of fragments and bTB positive status in the exploratory LOESS plot (Appendix A, Figure 2C), with every additional fragment linked to small increase in the odds of bTB breakdown (OR: 1.03; 95% CI: 1.02–1.05). Farm fragmentation was also entered as a categorical variable, and this model shown that in very highly fragmented farms (11+ fragments), the odds of bTB positive status was between 12% and 65% more likely than in farms with little fragmentation (2–4 fragments, OR: 1.36; 95% CI: 1.12–1.65), see Figure 2B. A positive trend between the odds of bTB positive status and fragmentation level was apparent. The test for linear trend reveals that the ordinal model was a poorer fit compared to the categorical model (Chi-Sq = 17.2, df = 3, *p* < 0.05).

BTB breakdown risk was also associated with fragment dispersal (OR: 1:001; 95% CI: 1.00–1.002 per 10 km), see LOESS plot in Appendix A, Figure 3E, however the effect sizes were very small (~0.1% increase in the probability of bTB breakdown per 10 km between fragments). When modelled as a categorical predictor, bTB breakdowns were between 19% and 55% more likely in farms with a “Very High” fragment dispersal (distances > 3.06 km between fragments) compared to farms with “Medium” levels of fragment dispersal (distances 0.53 km–1.38 km between fragments; OR: 1.35, 95% CI: 1.19–1.55); Figure 2C. There was also insufficient evidence that a model of linear trend was a better fit compared to a categorical model (Chi-Sq = 12.2, df = 2, *p* < 0.05).

The odds of bTB breakdown increased by between 6% and 9% for every additional km of shared boundary with neighbouring cattle farms (OR: 1.07, 95% CI: 1.06–1.09); see LOESS plot Appendix A, Figure 3G. The categorical model shows that the odds of bTB breakdown is elevated in farms with “Very High” levels of shared contact boundary (distances > 4.96 km) compared to medium levels (distances 1.49 km -2.84 km, OR: 1.72, 95% CI: 1.49–1.58), Figure 2D. The model of linear trend was a poorer fit to these data compared to the categorical model (Chi-Sq = 14.7, df = 2, *p* < 0.05). 

The number of bTB positive neighbours was also positively associated with the odds of bTB breakdown; for every additional neighbour, the odds increased by between 40% and 50% (OR: 1.45, 95% CI: 1.40–1.50), see LOESS plot Appendix A, Figure 3I. Likewise, compared to having no neighbours with bTB, having at least one bTB positive neighbour increased the odds of breakdown by between 100 and 141% (OR: 2.20, 95% CI: 2.01–2.41).

### 2.3. Confounding

It was not possible to build a model with each of the exposures included, in part because of the extent of correlation between explanatory variables, and also because such a model has very sparsely populated categories. Instead, we assessed how each of the factors of interested changed or “adjusted” the Odds Ratio (aOR) when the number of bTB positive neighbours was considered as a confounding factor. Confounding can make the relationship between an exposure and the outcome appear stronger or weaker than actual. We were particularly interested in whether the number of bTB positive neighbours was a positive confounder, which made the relationship between the factor of interest an bTB positive status appear stronger than it is, or whether it was a negative confounder making the relationship between the factor of interest an bTB positive status appear weaker. Positive confounding means that the aOR is closer to the null value (i.e., OR = 0) compared to the crude OR following the inclusion potential confounding factor, whereas negative confounding means that the aOR is further from the null value, compared to the crude OR. Table 3 shows the crude unadjusted ORs for univariable relationships compared with aORs for each of farm area, farm fragmentation, fragment dispersal and contact with neighbours after controlling for the presence of bTB positive neighbours. In the categorical model for farm area, it was observed that an association between breakdown risk and farm size was apparent in very large farms only, and so to simplify the analysis, a new variable for farm area was created, dichotomised at this point (>59.1 ha). Given that the OR for this variable was reduced by 22% once the number of bTB positive neighbours was taken into account, we considered the number of bTB positive neighbours as a positive confounder, making the positive relationship between farm area and bTB status appear stronger. Further investigation of this confounding revealed that the average very large farm had 1 bTB positive neighbour (IQR: 1–3, max = 14), compared to the average small farm (0 bTB positive neighbours, IQR 0–3, max = 5), see Figure 3A. Furthermore, 75% of very large farms had at least one bTB positive neighbour, compared to 37% of smaller farms. However, even after accounting for the number of bTB positive neighbours, very large farms were still positively associated with bTB breakdown risk, compared to smaller farms (aOR: 1.35, 95% CI: 1.20–1.52).

Confounding effects were also found between farm fragmentation and bTB positive neighbours; in very highly fragmented farms, the OR for bTB positivity was reduced by 46% once the number of infected neighbours was included in the model. Indeed, we observed up to 14 bTB positive neighbours in very highly fragmented farms, compared to a maximum of 5 in nonfragmented farms; Figure 3B. Some 84% of very highly fragmented farms had at least one bTB positive neighbour, compared to 45% of nonfragmented farms. After controlling for the presence of infected neighbours, there was no clear association between bTB breakdowns and different levels of farm fragmentation compared to the baseline (medium fragmentation).

The relationship between fragment dispersal and the odds of bTB breakdown was positively confounded with the number of bTB positive neighbours, Figure 3C. Some 42% of farms with low dispersal have at least one bTB positive neighbour, compared to 68% of very highly dispersed farms, and farms with very high dispersal were associated with elevated numbers of bTB positive neighbours. After controlling for the presence of infected neighbours, no clear associating remained between the odd of bTB breakdown and fragment dispersal.

Considerable confounding was also identified between neighbour contact category and the presence of bTB positive neighbours. After accounting for the number of infected neighbours, the OR for the category representing high contact between neighbours was reduced by 11%, whilst the OR of the category representing very high contact between neighbours was reduced by 41%. Thirty-six percent of farms with low contact metrics had at least one bTB positive neighbour, compared to 79% of farms with very high contact, and farms with very high contact metrics also had the largest number of bTB positive neighbours; see Figure 3D.

## 3. Discussion

Previous molecular studies from NI have revealed considerable spatial clustering of *M. bovis* genetic types in host populations, thereby affirming the important role of geographically localised processes in maintaining the bTB epidemic [35,36,37,38]. Although NI’s highly fragmented farmland is thought to be a contributing factor hampering eradication efforts, no studies to date have explored farm fragmentation as a risk factor for bTB, necessitating a deeper delve into the role of farm fragmentation as a facilitator of localised disease spread. Our principal finding is that after controlling for herd size, herd type, and spatial and temporal factors, increasingly fragmented farms were exposed to greater numbers of first-order spatial neighbours with bTB, which was directly associated with increases in bTB breakdown risk.

In our univariable models, we found that farm fragmentation, fragment dispersal and contact with neighbouring farms are each associated with increased odds of bTB breakdown, which is in concordance with previous studies. In the ROI, Byrne et al. (2020) observed an increase in bTB breakdown length of 6.6% in herds residing on farms with four parcels, compared to herds in farms with one parcel [28]. In the randomized badger culling trial (RBCT) area, Johnston et al. (2005) found that operating a farm over multiple premises was linked to a 79% increase in the odds of a bTB breakdown [39]. A later study in the same area also found that the number of contiguous holdings was an important risk factor for breakdowns [40]. In our study, these three variables of interest (farm fragmentation, fragment dispersal and contact with neighbouring farms) were, however, strongly confounded with the number of first-order spatial neighbours with bTB. Once the number of bTB positive neighbours was taken into account, the impact of each of these three factors on bTB breakdown risk was less clear. Our data therefore support the hypothesis that fragmentation metrics are largely proxy measures of exposure to neighbouring herds, and that the number of contiguous infected herds is therefore the main exposure of interest. Notwithstanding this, fragment dispersal could contribute to spreading disease from higher incidence areas to lower incidence areas via intra-herd movements. Indeed, a consequence of fragmentation is that parcels may be very widely dispersed; 25% of NI farms had a median distance of 3 km or more between fragments (to a maximum of 152 km) [27]. Unrecorded cattle movements between dispersed, but otherwise connected holdings could drive disease spread and intensify surveillance efforts [41]. However, the strong geographic clustering of *M. bovis* genetic types in NI, along with the relatively small median distances between fragments in the vast majority of farms, means that the importance of long range introductions is not evident. We do acknowledge that moving cattle between fragments could be a potential mechanism of disease spread within the herd, but because the distances involved are small, we argue that disease associated with intra-parcel movements between fragments should be practically indistinguishable from other short-range processes. Simulations of food-and-mouth spread between separate but associated premises in Scotland also suggest a diminished role for long-range intra-herd movements, but highlight the need to better understand land parcel occupancy to fully gauge the impact of intra-premises spread [42]. 

“Farm area” was the only variable where residual bTB breakdown risk remained after controlling for the presence of bTB positive neighbours; in very large farms (>59.1 ha) the odds of breakdown remained elevated by between 20% and 52% compared to smaller farms. This suggests that herds in very large farms experience additional risk beyond that presented by infection from neighbours. It may be that larger farms simply have more badgers on the land, as suggested by Vial et al. [15], or represent farms with mixed grazing of cattle and sheep, which has been linked to prolonged breakdowns in NI [43]. Additionally, the correlation between the variables used in this analysis means that larger farms are also more likely to be more fragmented [27], so the elevated risk linked to larger farms and more fragmented farms may instead represent additional positive confounding between these two exposures. 

The number of first-order spatial neighbours with bTB exhibited a particularly strong association with bTB breakdown risk; indeed, the presence of at least one neighbour with bTB was associated with more than double the odds of bTB breakdown compared to having no bTB positive neighbours. In some breakdown herds, up to 14 bTB positive first-order spatial neighbouring herds were identified. The importance of neighbourhood has been highlighted in an ROI study, where 35% of bTB animal incidence was associated with being within 1 km of other infected herds [22]. In a Northern Irish case-control study of 427 dairy herds, Denny and Wilesmith (1999) also found that case farms could be over twice as likely to have infected contiguous neighbours [32], and a study of 200 herds in the ROI found that the odds of an animal failing a tuberculin test increased by up for five-fold when a neighbouring farm was restricted in the prior six months [23]. Positive relationships between the odds of bTB breakdown and the number of confirmed breakdowns in neighbouring herds has also been found in UK cattle herds [16]. Whilst the presence of infected neighbours is therefore important in the UK and ROI, farm fragmentation is particularly prevalent in NI compared to other areas [29]. We therefore argue that herds in NI may be particularly exposed to disease originating from contiguous neighbours compared to infection from other sources, such as bought-in cattle [44]. Our study did not account for badger density or bTB prevalence, however, and it is highly likely that local wildlife (namely badgers, but possibly also deer) acts as an ongoing infection source within nearby herds, in the absence of any contiguous cattle transmissions. To overcome this limitation, we matched case and control herds to herds within the same geographic area (DVO); arguably case herds and control herds should therefore be exposed to similar risk from local badgers. We do acknowledge, however, that our approach could overlook within-DVO variation at smaller geographic scales.

Ultimately, tackling lateral spread of infection, requires understanding the drivers of fragmentation itself. Around 30% of NI’s farmland is dominated by the practice of informal short-term land-leasing called “conacre”, whereby a landowner may rent out only a portion of their farm for contracts <12 months. Leasing conacre is understood to be a pervasive mechanism driving increasing farm fragmentation, and the short-term nature of the lease means that land owners may be less likely to invest in conacre holdings, with subsequent poorer soil, reduced animal productivity and poorer biosecurity [45,46]. The conacre system is usually thought of as a factor impacting farm incomes and production, but we argue there may be an important role in conacre as an epidemiological driver of disease, leading to increased exposure between contiguous herds and potentially presenting with poorer biosecurity. Indeed, farmland fragmentation leads to substantial administrative and surveillance activities for staff within the Department for Agriculture, Environment and Rural Affairs (DAERA), who must identify neighbouring herds who may be at risk, extending beyond cattle residing on directly contiguous fragments at the time of breakdown; candidate neighbours for lateral tests include cattle grazed next to a breakdown herd in previous seasons, or who may graze next to the breakdown herd in future during the course of the breakdown. Consideration is also given to whether break-out cattle may spread infection to neighbouring herds, or whether there is contact with cattle along a laneway as cattle are herded between parcels. Risk of spread via possible wildlife-cattle contact or other indirect means is also considered. All at risk neighbours must be contacted (irrespective of biosecurity between parcels), and short interval full herd tests are carried out where risk of disease spread has been identified. The high levels of fragmentation and dispersal mean that large geographic extents may be involved, requiring local knowledge of the land and the people who farm it. 

### 3.1. Limitations

As this was a large scale computation study and not a field study, we were unable to accurately geo-reference herds within holdings. Potentially therefore, herds could occupy only one area within a very fragmented farm. However, a small study quantifying intra-herd cattle movements in NI shows that even fragmented land was grazed frequently, especially by beef animals, and that dairy animals in particular were frequently moved between pastures [47]. This suggests that even distal parcels on highly fragmented land may be utilised at least some of the time. Furthermore, as we had only GIS shapefiles of land parcel boundaries, and no information on whether the boundary was a bio-secure barrier (e.g., mature hedgerow), the level of contact between neighbouring farms is likely to be biased by overestimation. However, previous studies have found that between 67–79% of farm boundaries permit nose-to-nose contact [12,32], so whilst we acknowledge the measurement error, we posit that the impact on the conclusions are minimal. 

A limitation on determining the true extent of farms is introduced as a consequence of how the Basic Payment Scheme (BPS) system is administered. Either the land owner or tenant can claim for BPS and it is not possible to discern who is claiming for what parcels, and thus it is more challenging to be certain what land is being used and by whom. However, the BPS claimant is more often the farmer leasing and farming the land, and not the owner. This means that in practice, our estimations of farm area should largely reflect who is actively using the land parcels. 

One constraint to our conclusions surrounds the lateral testing process. Because bTB disclosure triggers an intensive epidemiological investigation into the herds and parcels surrounding a breakdown herd, surveillance efforts are not homogenous. Thus, there may be increased likelihood of disease being identified proximal to case herds, based on this factor alone. Whilst we hypothesise that high odds ratio associated with the presence of neighbours reflects the contiguous spread of disease; it could be debated that we are observing a consequence of enhanced surveillance and increased intensity of testing. We, however, argue against this interpretation; due to the annual testing schedule, bTB would be eventually detected in the herds surrounding control herds if it were present.

### 3.2. Future Work

The issue of fragmented farms poses an epidemiological risk for a wide range of pathogens in addition to bTB (e.g., BVD), however highly fragmented farms may also introduce logistical complexities into production, thereby decreasing the technical efficiency of dairy farms by [48]. Future work should therefore consider whether farm fragmentation in NI is a barrier to increased agricultural output. Furthermore, further studies to assess spatial patterns of conacre use in NI should be carried out, along with a quantification of the contribution of conacre to farm fragmentation. Tackling the bTB epidemic will require a more in-depth understanding of this highly localised phenomena, including the economic drivers and biosecurity implications of conacre land. There may also be various landscapes effects influencing transmission risk on-farm [39,49], if, for example, spillback risk may vary with land use heterogeneity.

## 4. Materials and Methods

We conducted a retrospective matched case-control observational study at the herd level to quantify the strength of association between the risk of bTB breakdown, and key metrics of farm fragmentation, fragment dispersal, and contact with contiguous farms. The study period ran between 1 January 2015–31 December 2017 inclusive. 

### 4.1. Study Region and Study Population

Northern Ireland (approximately 13,500 km^2^) is situated in the northeast of the island of Ireland, with a national herd of 1.6 million cattle distributed throughout approximately 20,000 farms. Approximately 13% of these holdings are dairy farms (n = 2600) and 70% are beef (n = 14,000), with a number of other herd types making up the remainder (e.g., breeding bulls) [50]. Cattle controls for *M. bovis* require that all bovines in NI over 42 days of age undergo annual testing using the Single Intradermal Comparative Cervical Tuberculin (SICCT) test, whereby SICCT-positive animals are removed from the herd and culled. Surveillance also involves the routine inspection of animal carcasses for lesions consistent with tuberculosis (LRS); suspect lesions are confirmed either way via histology or bacteriological investigations. We followed DAERA’s policy on bTB breakdown confirmation at the time [51], which requires meeting one of the following criteria: (1) a single animal with positive result to two of the following confirmatory tests (the SICCT test, abattoir inspection for lesions, histology, and bacteriology); or (2) the presence of five or more LRS animals during the course of a breakdown. 

### 4.2. Exposure Variables

#### 4.2.1. Herd Variables

Data on individual cattle, cattle herd demographics, and bTB tests and breakdowns were provided by the DAERA Animal and Public Health Information System (APHIS) [52]. APHIS variables for individual cattle herds were the median herd size for the calendar year, the year of bTB testing, the herd bTB status (positive herds had at least one confirmed breakdown in the calendar year) the Divisional Veterinary Office wherein the farm homestead was recorded (DVO), and the herd type (Table 1).

#### 4.2.2. Spatial Variables

Land parcel spatial boundaries for every land parcel associated with each cattle farm were made available from the DAERA Land Parcel Identification System, and used to derive model variables. The sum of all land parcels was used to derive the total farm area (ha). We also calculated the number of “fragments” associated with each cattle business, with a fragment defined as a spatially distinct functional unit of land parcels. This was more appropriate than using the total number land parcels, as all the land parcels within a fragment are epidemiologically linked, as farmers permit cattle to move between adjoining fields. Fragments were defined in a previous study [27], but briefly all parcels belonging to a single farm within 5 m of each other were aggregated together into a single unit. This distance is width of a narrow access road in NI, and therefore represents a meaningful bio-secure boundary. 

Fragment dispersal was defined using the median distance between fragments, calculated by measuring the Euclidian distance in km between fragment centroids. Contact with neighbouring farms was the length of shared perimeter between a cattle farm and its first-order spatial neighbours (Figure 4). As it is known that some land parcels may be used for noncattle activities, this was mitigated where possible by using Land Cover Map (LCM) data to determine which land parcels were potentially suitable for cattle farming (LCM classification 4 “improved grassland” and LCM classification 5 “neutral grassland”). This excludes land parcels classified as e.g., arable, woodland, bog, mountain or coast. Land Cover Maps for Northern Ireland for 2015 were purchased from the Centre for Ecology and Hydrology (https://www.ceh.ac.uk/, accessed on 6 December 2021) at 25 m^2^ resolution [53]. Farm fragmentation, fragment dispersal and contact with first-order spatial neighbours were recorded as continuous variables and recoded as categorical exposures; both continuous and categorical variables were considered in this analysis. Farm fragmentation was split into five categories, and fragment dispersal and fragment area were each split into four quartiles (Table 1). Lastly, for each herd, we derived the number contiguous herds in which a bTB breakdown was confirmed in the twelve months prior to breakdown. 

Whilst similar, each of farm fragmentation, fragment dispersal and contact with neighbouring farms represent different exposures. We argue that number of first-order spatial neighbours with bTB is the most specific measure of direct risk from contiguous herds, however we included the farm fragmentation variable as a separate measure as highly fragmented farms may be materially different from nonfragmented farms via management approaches related to grazing regime e.g., rotational grazing across multiple parcels. Calculating the length of shared boundary with neighbouring cattle farms is a more precise measure of contact with neighbouring herds, given that this metric was derived from boundaries where both internal and external parcels were potentially suitable for cattle farming. We also included fragment dispersal, as unrecorded, intra-herd cattle movements could lead to animals being moved from lower incidence areas to higher incidence areas, thereby increasing the risk of breakdown. In addition to this, widely dispersed land parcels may capture more landscape heterogeneity on-farm, with potentially better habitat provision for potential reservoir species via availability of rough pasture and woodland [54].

### 4.3. Participating Herds

Case herds were those with at least one confirmed bTB breakdown that started and ended during the dates of the study. For herds that had more than one confirmed breakdown, we included only the earliest breakdown (although 76.7% of herds experienced only a single breakdown). Eligible control herds were herds that remained bTB free throughout the study period. We matched on production type (as it was previously found that dairy herds tend to be associated with higher levels of farm fragmentation, and different production types may experience differential risk of bTB breakdown); herd size (as larger herds may reside on more fragmented land, are known to be at higher risk of bTB); and Divisional Veterinary Office (DVO), a local spatial variable associated with bTB administration (this variable is known to capture spatial variation in bTB risk and wildlife density), and year of bTB test. Herd size was the only noncategorical matching variable, and here we matched on herd size +/− 10 head of cattle. Cases and controls were matches in a 1:1 ratio on potential confounding variables; we used a higher ratio of controls to cases as it was not possible to match on all criteria.

### 4.4. Analysis

Descriptive statistics for each explanatory variable (counts and percentages for categorical variables; median and lower and upper Inter-Quartile Range (IQR) for continuous variables) were generated for case and control herds. Locally estimated scatterplot smoothing (LOESS) plots and boxplots were created for exploratory analysis. Correlations between explanatory variables were assessed using Spearman’ Rank Correlation Coefficient. Univariable and multivariable parameter estimates were derived using conditional GLMs via the clogit function in the *survival* package [55]. Pair IDs were used to control the matching. Analyses were carried out using both continuous and categorical explanatory variables. The Odds Ratios (OR’s) were reported, along with the 95% Lower and Upper Confidence Intervals (95% CI). The null hypothesis was that there is no association between the odds of bTB breakdown and each of the candidate exposures, after controlling for herd size, herd type, DVO and year of breakdown. Tests for linear trend (i.e., dose -response relationships) between bTB risk and categorical exposures were carried out by fitting models wherein the categories are treated as ordered factors as opposed to nominal categories. These simpler ordinal models were compared with the more complex categorical models using Likelihood Ratio Tests (LRTs). Here, the null hypothesis was that the simpler, ordinal model sufficiently fits the data. Furthermore, we hypothesise that that the number of bTB positive neighbours is likely to confound the relationship between bTB breakdown risk and each of farm area, farm fragmentation, fragment dispersal and contact with neighbouring farms. We considered confounding effects to be present where changes of 10% or more were observed in parameter estimates, after the infectious neighbours variable was added to the model [56]. We were particularly interested in any remaining association between the odds of bTB breakdown and each of farm area, farm fragmentation, fragment dispersal and contact with neighbouring farms, once the number of bTB positive neighbours was taken into account. This would suggest additional risk associated with these attributes beyond the presence of over the fence contact with infectious herds; here, we report the adjusted Odds Ratio (aOR) and 95% CI’s. The data were managed in MS SQL 2016 and analysis was carried out using R version 3.3.4. Figures were generated using *ggplot2* [57]. The full analytical procedure is presented in the R Markdown output (Appendix A). 

## 5. Conclusions

Northern Ireland’s agricultural system means that many farms are very fragmented, resulting in herds with high levels of direct contact with neighbouring farms, and sometimes considerable distances between fragments. We were ultimately interested in whether fragmentation was associated with odds of bTB breakdown, and whilst we found that farm area, farm fragmentation, fragment dispersal and contact with neighbouring farms were indeed associated with increased odds of breakdown, these factors were also confounded with the number of bTB positive neighbours. We found that highly fragmented farms were around twice as likely to have at least one bTB positive neighbour compared to nonfragmented farms, and that every bTB positive neighbour increased the odds of bTB breakdown by 40–50%. After accounting for the number of bTB positive neighbours in models of how bTB breakdown risk is associated with farm fragmentation metrics, we find no compelling evidence of a relationship between odds of bTB breakdown and fragmentation. This signifies that these farm fragmentation metrics are probably capturing the effects of “over the fence” cattle-to-cattle transmission. Although it is possible that similar observations arise if herds in the area were exposed to a shared infection source such as infected wildlife, our geographical matching criteria make this less likely. Tackling local spread will require a deeper understanding of the patterns, drivers and characteristics of NI’s conacre rental system, which is a key factor influencing land fragmentation.

## Figures and Tables

**Figure 1 pathogens-11-00299-f001:**
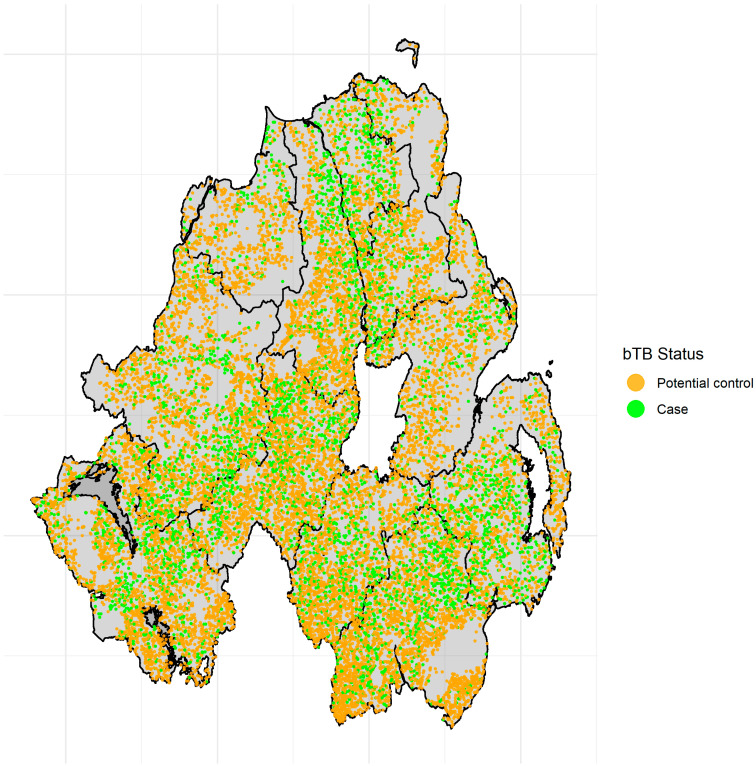
Spatial distribution of case herds and all potential control herds across the 10 DVO areas in NI.

**Figure 2 pathogens-11-00299-f002:**
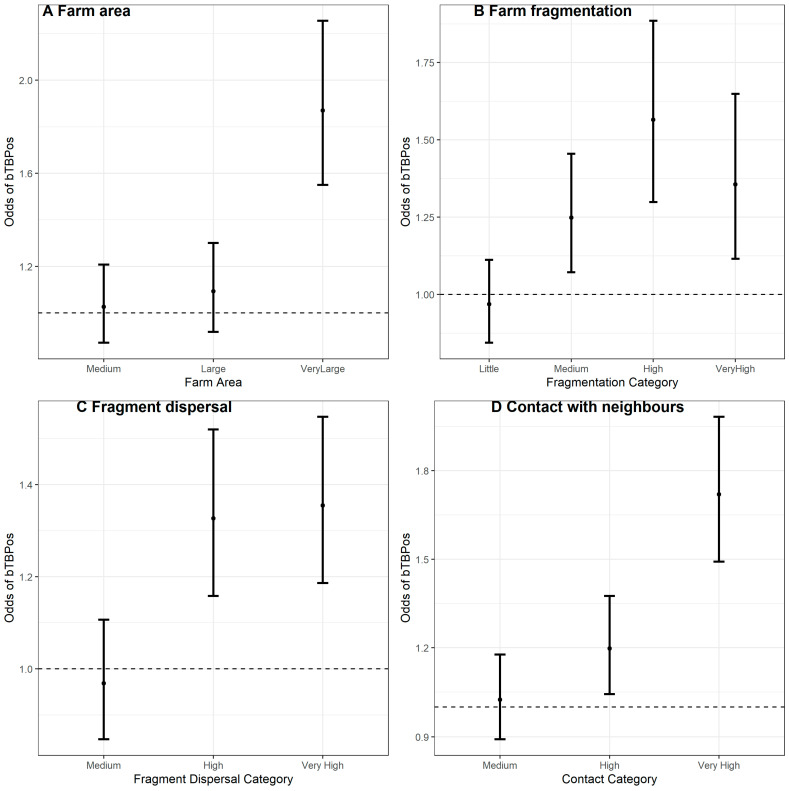
Plots of OR’s and 95% CI for each factor in the four categorical predictors; (**A**) Farm area, (**B**) Farm fragmentation, (**C**) Fragment dispersal, and (**D**) Contact with neighbours.

**Figure 3 pathogens-11-00299-f003:**
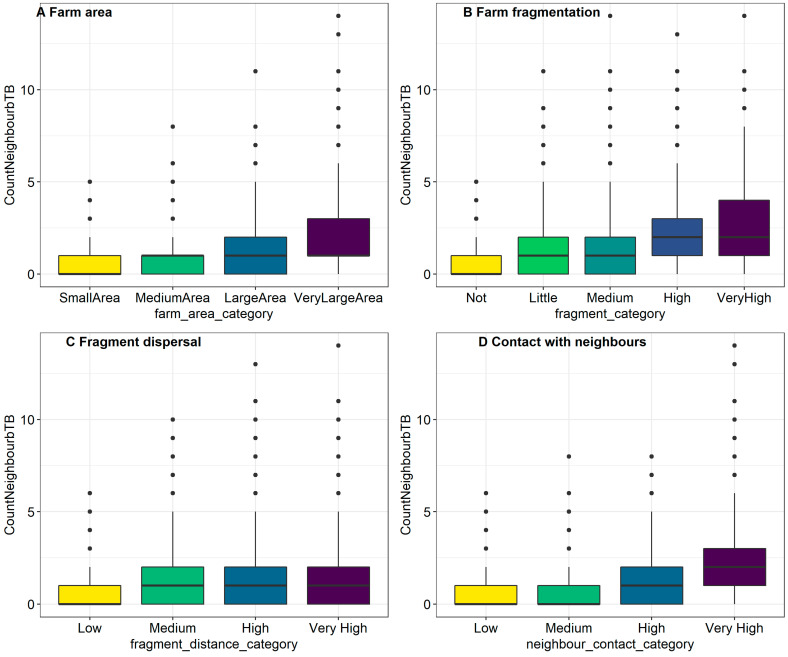
Illustration of confounding between the number of bTB positive neighbours and each of (**A**) farm area, (**B**) farm fragmentation, (**C**) fragment dispersal, and (**D**) contact with neighbouring farms.

**Figure 4 pathogens-11-00299-f004:**
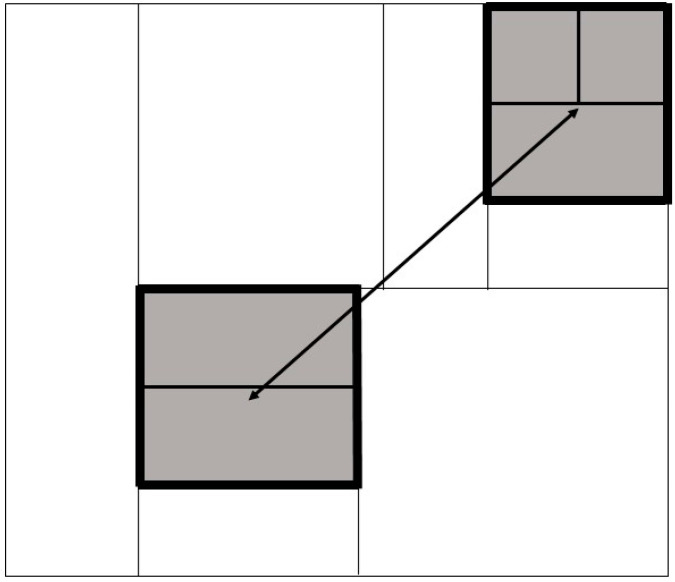
A schematic of definitions used in this manuscript. In this example, the index farm is shown in grey, surrounded by land parcels belonging to first-order spatial neighbours in white. The index farm has five land-parcels distributed across two fragments. The distance between the fragment centroids (in km) is shown by the double headed arrow. The contact with neighbouring farms (in km) is the perimeter of the fragments, coloured with the thickest black line.

**Table 1 pathogens-11-00299-t001:** The distribution of each explanatory variable in the cases, control sample, and whole cattle population.

		bTB Status	
	Negative, N = 4637	Positive, N = 4637	Population, N = 570,241
Year			
2015	1774 (38%)	1774 (38%)	19,008 (33%)
2016	1389 (30%)	1389 (30%)	19,008 (33%)
2017	1474 (32%)	1474 (32%)	19,008 (33%)
DVO			
Armagh	428 (9.2%)	428 (9.2%)	5487 (9.6%)
Ballymena	269 (5.8%)	269 (5.8%)	3465 (6.1%)
Coleraine	590 (13%)	590 (13%)	6450 (11%)
Dungannon	590 (13%)	590 (13%)	6954 (12%)
Enniskillen	566 (12%)	566 (12%)	7848 (14%)
Larne	226 (4.9%)	226 (4.9%)	3882 (6.8%)
Londonderry	119 (2.6%)	119 (2.6%)	2061 (3.6%)
Newry	688 (15%)	688 (15%)	8934 (16%)
Newtownards	515 (11%)	515 (11%)	4458 (7.8%)
Omagh	646 (14%)	646 (14%)	7470 (13%)
NA	-	-	15 (<0.1%)
Herd Type			
Breeder	1251 (27%)	1251 (27%)	28,415 (50%)
Dairy	1267 (27%)	1267 (27%)	8128 (14%)
Finisher	690 (15%)	690 (15%)	5976 (10%)
Other	1429 (31%)	1429 (31%)	14,505 (25%)
Median herd size	81 (41, 156)	83 (40, 173)	40 (19, 89)
Total farm area (ha)	42 (26, 72)	50 (27, 84)	31 (16, 59)
Farm area category ^1^			
Small (1.2 ha–16.4 ha)	583 (13%)	538 (12%)	14,164 (25%)
Medium (16.41 ha–31.2 ha)	938 (20%)	852 (18%)	14,353 (25%)
Large (31.21 ha–59.1 ha)	1570 (34%)	1343 (29%)	14,255 (25%)
Very Large (>59.1 ha)	1546 (33%)	1904 (41%)	14,252 (25%)
N fields	28 (17, 44)	33 (20, 51)	24 (14, 39)
N fragments	4 (2, 6)	4 (3, 7)	3 (2, 6)
Fragmentation category ^2^			
Not_fragmented (1 fragment)	526 (11%)	488 (11%)	4691 (8.2%)
Little fragmentation (2–4 fragments)	2226 (48%)	1974 (43%)	27,164 (48%)
Medium_fragmentation (5–7 fragments)	1062 (23%)	1165 (25%)	12,353 (22%)
High_fragmentation (8–10 fragments)	415 (8.9%)	551 (12%)	9476 (17%)
Very_high_fragmentation (11+ fragments)	408 (8.8%)	459 (9.9%)	3340 (5.9%)
Median distance fragments (km)	1.52 (0.72, 3.23)	1.83 (0.86, 3.57)	1.38 (0.52, 3.05)
Fragment distance category ^3^			
Low	798 (17%)	697 (15%)	14,255 (25%)
Medium	1338 (29%)	1116 (24%)	14,256 (25%)
High	1282 (28%)	1425 (31%)	14,256 (25%)
Very High	1219 (26%)	1399 (30%)	14,255 (25%)
Total shared boundary (km)	3.4 (2.0, 5.7)	4.1 (2.2, 6.7)	
Neighbour contact category ^4^			
Low (0 km–0.52 km)	763 (16%)	644 (14%)	14,256 (25%)
Medium (0.53 km–1.38 km)	1086 (23%)	918 (20%)	14,256 (25%)
High (1.39 km–3.05 km)	1255 (27%)	1181 (25%)	14,256 (25%)
Very High (>3.06 km)	1533 (33%)	1894 (41%)	14,256 (25%)
Count neighbours bTB	1 (0, 1)	1 (0, 3)	0 (0, 1)
Any neighbour bTB ^5^	2463 (53%)	3284 (71%)	27,074 (48%)

The data were matched on variables Year, DVO, Herd Type and Median herd size. Counts (N) and percentages (%) are reported for categorical variables, and medians with IQR’s are reported for continuous variables. ^1^ Total farm area (ha) categorised; ^2^ N fragments categorized; ^3^ Median distance fragments (km) categorised; ^4^ Total shared boundary (km) categorised; ^5^ Count neighbours bTB dichotomised.

**Table 2 pathogens-11-00299-t002:** Results of the univariable analysis for both categorical and continuous predictors.

	**Categorical**			**Continuous**	
**Variable**	**OR**	**95% CI**		**OR**	**95% CI**
**Farm area category**			Total farm area (per ha)	1.00	1.00, 1.00
Small (1.2 ha–16.4 ha)	—	—			
Medium (16.41 ha–31.2 ha)	1.03	0.87, 1.21			
Large (31.21 ha–59.1 ha)	1.09	0.92, 1.30			
Very Large (>59.1 ha)	1.87	1.55, 2.25			
**Fragment category**			N fragments (per fragment)	1.03	1.02, 1.05
Not_fragmented (1 fragment)	—	—			
Little fragmentation (2–4 fragments)	0.97	0.84, 1.11			
Medium_fragmentation (5–7 fragments)	1.25	1.07, 1.46			
High_fragmentation (8–10 fragments)	1.56	1.30, 1.88			
Very_high_fragmentation (11+ fragments)	1.36	1.12, 1.65			
**Fragment distance category**			Median distance fragments (per 10 km) ^1^	1.00	1.00, 1.00
Low (0 km–0.52 km)	—	—			
Medium (0.53 km–1.38 km)	0.97	0.85, 1.11			
High (1.39 km–3.05 km)	1.33	1.16, 1.52			
Very High (>3.06 km)	1.35	1.19, 1.55			
**Neighbour contact category**			Total shared boundary (per km)	1.07	1.06, 1.09
Low (0 km–1.48 km)	—	—			
Medium (1.49 km–2.84 km)	1.02	0.89, 1.18			
High (2.85 km–4.95 km)	1.20	1.04, 1.38			
Very High (>4.96 km)	1.72	1.49, 1.98			
**Any neighbour bTB**	2.20	2.01, 2.41	Count neighbours bTB	1.45	1.40, 1.50

For continuous predictors, the Odds Ratio (OR) and 95% lower and upper confidence intervals (95% CI) are shown. For categorical predictors, the OR and 95% CI are also shown, with reference to the baseline category. ^1^ Despite scaling from 1 km to 10 km the model coefficient is still very small.

**Table 3 pathogens-11-00299-t003:** The crude ORs and adjusted ORs showing how the number of neighbours with bTB confounds the relationship between each of farm area, farm fragmentation, fragment dispersal and contact with neighbouring farms.

Model	Variable	Unadjusted Analysis		Adjusted Analysis	
		OR	95% CI	aOR	95% CI
Farm area	Farm area category ^1^				
	Small/Medium/Large (≤59.1 ha)	—	—	—	—
	Very Large (>59.1 ha) ^†^	1.74	1.56, 1.95	1.35	1.20, 1.52
	Count neighbours bTB (per neighbour)	1.45	1.40, 1.50	1.43	1.38, 1.48
Fragmentation	Fragment category				
	Not_fragmented (1 fragment)	—	—	—	—
	Little fragmentation (2–4 fragments) ^†^	0.97	0.84, 1.11	0.83	0.72, 0.96
	Medium fragmentation (5–7 fragments)	1.25	1.07, 1.46	0.96	0.81, 1.12
	High fragmentation (8–10 fragments)	1.56	1.30, 1.88	1.06	0.87, 1.30
	Very high fragmentation (11+ fragments) ^†^	1.36	1.12, 1.65	0.73	0.59, 0.91
	Count neighbours bTB (per neighbour)	1.45	1.40, 1.50	1.45	1.40, 1.51
Fragment dispersal	Fragment distance category				
	Low (0 km–0.52 km)	—	—	—	—
	Medium (0.53 km–1.38 km) ^†^	0.97	0.85, 1.11	0.84	0.73, 0.97
	High (1.39 km–3.05 km)	1.33	1.16, 1.52	1.01	0.87, 1.17
	Very High (>3.06 km)	1.35	1.19, 1.55	1.01	0.88, 1.16
	Count neighbours bTB (per neighbour)	1.45	1.40, 1.50	1.44	1.39, 1.49
Neighbour contact	Neighbour contact category				
	Low (0 km–1.48 km)	—	—	—	—
	Medium (1.49 km–2.84 km)	1.02	0.89, 1.18	0.91	0.79, 1.05
	High (2.85 km–4.95 km)	1.20	1.04, 1.38	0.95	0.82, 1.10
	Very High (>4.96 km)	1.72	1.49, 1.98	1.02	0.87, 1.19
	Count neighbours bTB (per neighbour)	1.45	1.40, 1.50	1.44	1.39, 1.49

^1^ Dichotomised at “Very large farms”; ^†^ Levels of a variable which were significantly different from the baseline category, after controlling for the number of contiguous neighbours.

## Data Availability

An anonymised version of the dataset used in this analysis is available upon request from the corrosponding author.

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
