# Peer review of "Investigating Farm Fragmentation as a Risk Factor for Bovine Tuberculosis in Cattle Herds: A Matched Case-Control Study from Northern Ireland"

_pathogens, 2022, doi:10.3390/pathogens11030299_

Round 1

Reviewer 1 Report

Whilst the findings of this study are not altogether surprising, they are nonetheless important to publish. The paper has a clearly presented background and aims of the study, however, please check the reference numbers, especially from no. 30 onwards as these are almost certainly incorrect.

It is not until Table 2 that the definition of the categories, such as farm area, are described. These should be described along with Table 1 so that Table 1 can be understood on its own.

Furthermore, the boundaries used for categorising variables, such as farm area, should be justified. When were these set, before the analysis? Unless this is described clearly, the analysis could be open to the charge of being biased.

The first paragraph of section 2.3 relating to confounding is hard to follow. Please consider rephrasing to make it easier for someone unaccustomed with confounding to follow. Related to that I suggest highlighting in Table 3 those Odds Ratios that went from significant to not as result of their adjustment to take account of the number of neighbours with bTB.

Please improve the legibility of text in Figure 3 and expand on the legend to make the figure easier to understand without reference to the body of the paper.

For the discussion regarding the association between farm area and bTB breakdown risk, what can be said regarding the association between farm area and number of cattle, and how the latter might influence the risk of breakdown risk. Was herd size considered in this respect?

As the association between the odds of bTB breakdown and the number of infected contiguous neighbours does not necessarily imply “over the fence” infection but could point to a shared environmental source of infection, it was good to see the authors discussed this point (lines 150-156, and Conclusions). However, is there any way for the authors to know, of the contiguous infected farms, where proper biosecurity (double fencing) was being applied and where it was not, especially as they imply this may be less likely in situations of conacre? This may help tease apart this issue, although I see they have partly addressed this as a limitation of their study.

Page numbering was inconsistent and line numbering only began from the top of page 5. The following typos were identified:

Line 17 – ‘for’ should be ‘fit’?

Line 21 – ‘Figure 3H’ should be ‘Figure 2H’.

Line 28 – ‘Figure 3I’ should be ‘Figure 2I’.

Check M. bovis is italicised throughout (not in lines 83, 113, 233).

Line 281 – should be Figure 4? If so, is the reference to Figure 1 at line 267, correct, or does it refer to this figure (4)? Also, the shading referred to in this figure appears to have been lost.

Line 378 – spelling of anonymised.

Details of any software used for analysis should be properly referenced in Section 4.4.

Author Response

Dear editors and reviewers,

We thank you for taking the time to go review our manuscript and offer comments for improvement.  Please see below how each item has been dealt with,

On behalf of all the authors,

Dr Georgina Milne

REVIEWER 1

We wish to thank R1 for their helpful and constructive comments.  We hope that the following addresses the points raised:

  1. please check the reference numbers, especially from no. 30 onwards as these are almost certainly incorrect.

Response: completed – this arose after moving the M&M section to the end.

  1. It is not until Table 2 that the definition of the categories, such as farm area, are described. These should be described along with Table 1 so that Table 1 can be understood on its own.

Response: completed

  1. The first paragraph of section 2.3 relating to confounding is hard to follow. Please consider rephrasing to make it easier for someone unaccustomed with confounding to follow.

  1. Response: completed

  1. Related to that I suggest highlighting in Table 3 those Odds Ratios that went from significant to not as result of their adjustment to take account of the number of neighbours with bTB.

Response: completed

  1. Please improve the legibility of text in Figure 3 and expand on the legend to make the figure easier to understand without reference to the body of the paper.

Response: completed

  1. For the discussion regarding the association between farm area and bTB breakdown risk, what can be said regarding the association between farm area and number of cattle, and how the latter might influence the risk of breakdown risk. Was herd size considered in this respect?

Response: We wanted to eliminate the “herd size” effect, so for these analysis, herds were matched on herd size prior to analysis.  If we hadn’t matched on herd size we would expect herd size to be an important variable in the model.

  1. As the association between the odds of bTB breakdown and the number of infected contiguous neighbours does not necessarily imply “over the fence” infection but could point to a shared environmental source of infection, it was good to see the authors discussed this point (lines 150-156, and Conclusions). However, is there any way for the authors to know, of the contiguous infected farms, where proper biosecurity (double fencing) was being applied and where it was not, especially as they imply this may be less likely in situations of conacre? This may help tease apart this issue, although I see they have partly addressed this as a limitation of their study.

Response: The reviewer is correct in that having knowledge of the local badger density and/or infection prevalence “risk” maps would be useful on teasing out over the fence v wildlife mediated infection.  These data are not available, which is unfortunate.  However, in a previous publication by some of same authors, DVO was found to correlate with badger density (Milne G, Allen A, Graham J, Lahuerta-Marin A, McCormick C, Presho E, Reid N, Skuce R, Byrne AW. 2020. Bovine tuberculosis breakdown duration in cattle herds: an investigation of herd, host, pathogen and wildlife risk factors. PeerJ 8:e8319 https://doi.org/10.7717/peerj.8319).  We therefore elected to match on DVO as explained to try and capture some local wildlife effects.

Regarding the second point – we agree that the information on the quality of land parcel boundaries would help refine these conclusions, along with some estimate of which parcels are in use, and when.  We are looking at exploring this further for future biosecurity studies, as e.g. satellite and LIDAR maps are becoming more available.     

  1. Page numbering was inconsistent and line numbering only began from the top of page 5.

Response: Page numbers amended by editing team, line numbers now removed

  1. Line 17 – ‘for’ should be ‘fit’?

Response: completed

  1. Line 21 – ‘Figure 3H’ should be ‘Figure 2H’.

Response: completed

  1. Line 28 – ‘Figure 3I’ should be ‘Figure 2I’.

Response: completed

  1. Check M. bovis is italicised throughout (not in lines 83, 113, 233).

Response: completed

  1. Line 281 – should be Figure 4? If so, is the reference to Figure 1 at line 267, correct, or does it refer to this figure (4)?

Response: completed

  1. Also, the shading referred to in this figure appears to have been lost.

Response: completed

  1. Line 378 – spelling of anonymised.

Response: completed

  1. Details of any software used for analysis should be properly referenced in Section 4.4.

Response: completed

Reviewer 2 Report

Brief summary:

The manuscript deals with a current and very serious problem that affects a lot of countries that are fighting bTB.

The authors have carried out a case-control matched study to quantify the risks of bTB breakdown with farm area, farm fragmentation, fragment dispersal, and contact with neighboring herds.

The results of the study show small but significant increases in the risk of bTB breakdown associated with each farm fragmentation, farm area, fragment dispersal, and contact with neighboring herds; however, these relationships were strongly confounded with the number of contiguous neighbors with bTB.

The authors have demonstrated that each bTB infected neighbor led to a 40% to 50% increase in the odds of breakdown and that highly fragmented farms were nearly twice as likely to have a bTB-positive neighbor as non-fragmented farms.

These results suggest that herd fragmentation increasingly exposes herds to infection originating from first-order spatial neighbors. Thus, contiguous spread between herds is a particularly important component of the spread of bTB in Northern Ireland in particular due to Northern Ireland’s agricultural landscape and highly fragmented farming structures.

This topic is of particular interest because the described problem is also present in other European Countries. Therefore, the study could be useful to help government agencies understand the relevance of specific bTB controls for these regions and territories, and ad hoc bTB containment measures.

Broad comments:

The manuscript deals with a current and very serious problem that affects a lot of countries that are fighting bTB.

The reading of the manuscript is very fluent and pleasant, the only aspect to be reported is related to the use of abbreviations. It is important to introduce the abbreviation the first time a term is used, even in the abstract (where it would be better not to use abbreviations).

E.g. NI in the abstract, or some technical terms such as DAERA, that are described in the Materials and Methods section but not in the previous paragraphs. Probably this is due to the circumstance that initially the Materials and Methods section was not at the end of the article, however just pay attention to all the abbreviations in the text and in the captions.

Figures

In Figure 1 it is really difficult to distinguish the bTB status Potential control vs case.

In Figure 3 it is difficult to read the description of the different types of the studied areas (A, B, C, D).

Figure 4 is really hard to understand, maybe it could become supplementary material?

Author Response

We wish to thank R2 for their helpful and constructive comments.  We hope that the following addresses the points raised:

  1. The reading of the manuscript is very fluent and pleasant, the only aspect to be reported is related to the use of abbreviations. It is important to introduce the abbreviation the first time a term is used, even in the abstract (where it would be better not to use abbreviations).
    Response: completed – we have went through the manuscript and ensured that all abbreviations are introduced, and also removed abbreviations from the abstract.
  1. In Figure 1 it is really difficult to distinguish the bTB status Potential control vs case.
    Response: Colours changes in figure
  1. In Figure 3 it is difficult to read the description of the different types of the studied areas (A, B, C, D).
    Response: completed
  2. Figure 4 is really hard to understand, maybe it could become supplementary material
    Response: Figure 4 has been updated with improved shading; it should be easier to understand now.

Reviewer 3 Report

This is a really interesting paper, adding some new information on farm related factors in bTB transmission in NI. The authors rightly acknowledge some of the limitations of the study in their discussion, but it is still of interest and relevance to bTB control.

Page 1

Abstract          4th line: does “……highly fragmented farming structures….” make sense? Could just be “Northern Ireland’s agricultural landscape is comprised of highly fragmented farms, distributed across spatially discontinuous land parcels, and these are thought to facilitate localised spread.”

                        8th line: could say ‘each factor’ rather than listing them all again

Page 2

Introduction    3rd paragraph: is ‘and fencing off badger latrines’, whilst clearly a good idea, relevant to contiguous cattle-to-cattle spread or indeed to this paper?

Materials and methods should be here rather than on p12 and subsequent section numbering is therefore incorrect!

Results

2.1 Descriptive results

1st line: The final dataset consisted of 19,008 herds, from which 4,637 (24.4%) has at least one confirmed bTB breakdown during the three year study period - this should be ‘….…had at least….’?

3rd line: These case herds were matched to 4,637 controls who did not have a bTB breakdown -  this should be ‘…..that did not have……..’?

Page 3

Table 1

It would make reading easier if ‘Farm area category’, ‘Fragmentation category’, ‘Fragmentation distance category’ and ‘Neighbour contact category’ were better defined either in the table itself or in the footnote - this has been done in Table 2. An alternative would have been to tabulate the definitions of all categories separately.

Page 5

I appreciate this may be the publication’s style, but these charts look very squashed on the page and might benefit from borders to make individual charts and their headings clearer.

Page 6           

line 9: as this relates to the table should it be left justified (as for Table 1)?

line 21: should this be referring to Figure 2H

line 23: “Medium” levels

line 28: should this be referring to Figure 2I

Page 8

Again, the layout seems quite ‘squashed’. The chart headings (A, B, C, D) would benefit from being a larger font size.

Page 9

Table 3: less essential in this table compared to Table 1, but the categories could be defined again for consistency

line 80: footnote needs to be left justified (as for Table 1)

line 83: M. bovis (italics)

Page 10

line 113 M. bovis (italics)

line 138 is there an extra space typo between ‘identified’ and ‘The’?

Page 11

line 151 Maybe ‘……likely that wildlife may act…….’ as you don’t know what wildlife if any is there or its infection status. If you mean badgers (which you probably do), then say badgers (not ‘wildlife’).

line 174: again ‘badgers’ rather than ‘wildlife’? or are you considering other species? Do you mean direct (which is currently implied) or indirect contact with badgers (much more likely)?

Page 12

Materials and methods is in the wrong place

line 233 M. bovis (italics)

Pager 14

Figure 2           ‘light grey’ is not so evident in my copy. Could go for another colour

line 292           is there an extra space typo between ‘of’ and ‘shared’?

Page 16

References:     Numbering throughout the references is duplicated

                        The first reference has been duplicated

Author Response

REVIEWER 3

We wish to thank R3 for their helpful and constructive comments.  We hope that the following addresses the points raised:

  1. Abstract 4th line: does “……highly fragmented farming structures….” make sense? Could just be “Northern Ireland’s agricultural landscape is comprised of highly fragmented farms, distributed across spatially discontinuous land parcels, and these are thought to facilitate localised spread.

Response: completed

  1. 8th line: could say ‘each factor’ rather than listing them all again

Response: completed

  1. Introduction    3rd paragraph: is ‘and fencing off badger latrines’, whilst clearly a good idea, relevant to contiguous cattle-to-cattle spread or indeed to this paper?

Response: well spotted – this has been removed

  1. Materials and methods should be here rather than on p12 and subsequent section numbering is therefore incorrect!

Response: we are following the journal template where M&M comes at the end

  1. 1st line: The final dataset consisted of 19,008 herds, from which 4,637 (24.4%) has at least one confirmed bTB breakdown during the three year study period - this should be ‘….…had at least….’?

Response: completed

  1. 3rd line: These case herds were matched to 4,637 controls who did not have a bTB breakdown - this should be ‘…..that did not have……..’?

Response: completed

  1. It would make reading easier if ‘Farm area category’, ‘Fragmentation category’, ‘Fragmentation distance category’ and ‘Neighbour contact category’ were better defined either in the table itself or in the footnote - this has been done in Table 2. An alternative would have been to tabulate the definitions of all categories separately.

Response: completed

  1. I appreciate this may be the publication’s style, but these charts look very squashed on the page and might benefit from borders to make individual charts and their headings clearer.

Response: we have removed some non-essential panels of this figure, and refer to supplementary material instead

  1. line 9: as this relates to the table should it be left justified (as for Table 1)

Response: completed

  1. line 21: should this be referring to Figure 2H

Response: completed

  1. line 23: “Medium” levels

Response: completed

  1. line 28: should this be referring to Figure 2I

Response: completed

  1. Again, the layout seems quite ‘squashed’. The chart headings (A, B, C, D) would benefit from being a larger font size.

  1. Table 3: less essential in this table compared to Table 1, but the categories could be defined again for consistency

Response: completed

  1. line 80: footnote needs to be left justified (as for Table 1)

Response: completed

  1. line 83: bovis (italics)

Response: completed

  1. line 138 is there an extra space typo between ‘identified’ and ‘The’?

Response: completed

  1. line 151 Maybe ‘……likely that wildlife may act…….’ as you don’t know what wildlife if any is there or its infection status. If you mean badgers (which you probably do), then say badgers (not ‘wildlife’).

Response: completed

  1. line 174: again ‘badgers’ rather than ‘wildlife’? or are you considering other species? Do you mean direct (which is currently implied) or indirect contact with badgers (much more likely)?

Response: we have changed the text to mention badgers and possibly deer.  We agree that indirect contact is generally the most likely mechanism of transmission.  I cannot find where we’ve mentioned that direct badger-cattle transmission is more likely - we mention both in the introduction but don’t dwell on comparing direct/indirect. 

  1. Materials and methods is in the wrong place

Response: The template for Pathogens shows that M&M should be in section four, at the end pf the manuscript

  1. line 233 bovis (italics)

Response: completed

  1. ‘light grey’ is not so evident in my copy. Could go for another colour

Response: completed

  1. Is there an extra space typo between ‘of’ and ‘shared’?

Response: completed

  1. Numbering throughout the references is duplicated

Response: completed

  1. The first reference has been duplicated

Response: completed